# CBCT Assessment of Gubernacular Canals on Permanent Tooth Eruption in Down’s Syndrome

**DOI:** 10.3390/jcm12103420

**Published:** 2023-05-12

**Authors:** Carlos Eduardo Vieira da Silva Gomes, Athus Michel Flexa Conceição, Sérgio de Melo Alves Júnior, Ricardo Roberto de Souza Fonseca, Rogério Valois Laurentino, Luiz Fernando Almeida Machado

**Affiliations:** 1School of Dentistry, Federal University of Pará, Belém 66075-110, PA, Brazil; dr.carlosgomes@hotmail.com (C.E.V.d.S.G.); sergiomalves@gmail.com (S.d.M.A.J.); 2Biology of Infectious and Parasitic Agents Post-Graduate Program, Federal University of Pará, Belém 66075-110, PA, Brazil; ricardofonseca285@gmail.com (R.R.d.S.F.);; 3Virology Laboratory, Institute of Biological Sciences, Federal University of Pará, Belém 66075-110, PA, Brazil

**Keywords:** oral and systemic disease interactions, special care dentistry, cone-beam computed tomography, down syndrome

## Abstract

Background: The gubernacular canal (GC) is an important dental structure that enables the alveolar bone ridge cohesion of permanent teeth, although GC absence may indicate a dental eruption that might be associated with certain syndromes such as Down’s syndrome. This study aims to correlate the eruptive delay of permanent teeth in individuals with Down’s syndrome (Ds) and the gubernacular canal (GC) through cone-beam computed tomography (CBCT). Methods and Results: This cross-sectional study was conducted between January and July 2022 with a total of 31 individuals (G1 = 16 nonsyndromic and G2 = 15 Down’s syndrome) who went through imaging evaluation using CBCT with the following acquisition parameters: tube voltage of 95 kVp, tube current of 7 mA, exposure time of 5.9 s and voxel sizes and field of view 0.15 mm and 0.30 mm, respectively. The imaging evaluation was to assess whether all teeth analyzed had the presence of GC and/or teeth eruption disturbance, with a descriptive statistical analysis of relative frequencies and quantitative variables as well as the *p*-value (*p* < 0.005) by G Test. Results: A total of 618 teeth among 31 individuals were analyzed, 475 (76.8%) GC were detected by CBCT in 23/31 patients and of these, 6 belonged to G2. G2 had a decreased GC detection rate (*n* = 180–37.9%) and the most common tooth with GC detected was the mandibular 1st molar (21 GC/25 teeth—84%) and the absence of GC was most frequently observed in impacted and delayed/unerupted teeth of Ds individuals. Conclusion: We concluded that GC absence was higher among Ds individuals, explaining the increased rates of unerupted or impacted teeth in Ds individuals.

## 1. Introduction

Mixed dentition is a stage of growth and development that begins with the dental eruption of the first permanent teeth at about 6 years old and continues until 12 years old [1,2,3]. In this stage, the oral cavity undergoes certain physiologic changes including deciduous teeth root resorption followed by exfoliation of permanent teeth within the gubernaculum dentis, which is made by the gubernacular cord, after which, osteoblastic activity forms the gubernacular canal, which is important in the eruption process, representing an eruption path through maxillary bone [4,5].

As mentioned previously, both the gubernacular canal and cord are intraoral structures very important to the tooth-eruption process, although a small number of studies in the literature about GC and cord demonstrated their influence in permanent teeth eruption physiologic processes, especially in the follicular theory [5,6]. The gubernacular cord is a structure composed of conjunctive tissue which connects the permanent tooth follicle to the gingiva and is located in the maxillary alveolar ridge below deciduous teeth guiding the course of permanent teeth eruption [7].

The gubernacular cord formation begins with the remnants of cells from the dental lamina that are assembled as a fibrous cord through the enamel organ’s reduced epithelium toward the oral mucosa [8,9,10,11,12,13]. In the gubernacular cord, there are chemical inflammatory mediators, epithelial growth factors, osteoclasts, and osteoblasts, and during the bone demineralization process, a space is formed around the gubernacular cord, which is denominated the gubernacular canal (GC) [8,9,10,11,12,13]. However, in some cases, the physiological permanent teeth eruption process is delayed due to GC obstruction.

GC obstruction might occur for various reasons such as impacted teeth, abnormal teeth orientation, maxillary atresia, overly dense bone, excessively soft tissue, craniofacial abnormalities, genetic abnormalities such as Down’s syndrome (DS), supernumerary teeth, and odontogenic tumors such as odontoma [5,14]. Ds, also known as trisomy 21, is a genetic disorder caused by the presence of an extra part of a third copy of chromosome 21. The phenotypic characteristics of individuals with Ds have some specific craniofacial and oral cavity features which might include brachycephaly, a low nasal bridge, a small maxilla, an underdeveloped midfacial region, tongue size, and position alterations, increased rates of periodontal disease, class III malocclusion, anterior open bite, fissured tongue, dental anomalies, and delayed dental eruption [15,16,17,18].

GC obstruction among individuals with Ds has not yet been reported in the literature. A few studies demonstrated oral cavity abnormalities in individuals with Ds; however, this does not correlate with GC obstruction. De Moraes et al. [15] evaluated the incidence of dental anomalies in permanent dentition among individuals with Ds using panoramic X-rays and, as a result, the authors observed dental anomalies in 50.47% of the cases and retained teeth or supernumerary teeth were among the most prevalent abnormalities.

Cuoghi et al. [16] evaluated the prevalence of dental anomalies in Brazilian individuals with Ds, and according to de Moraes et al. [15], the authors also used panoramic X-rays and there was a high prevalence of impacted teeth within dental anomalies. The relationship between Ds and GC obstruction characteristics may remain unclear due to the difficulties in accessing the area mainly because of the lack of accurate imaging technology. However, in recent years, the advances in cone-beam computed tomography (CBCT) have possibly made this a valuable tool in the diagnosis of GC obstruction and orthodontic, nonsurgical, and surgical treatment planning of individuals with Ds. This study aimed to correlate the eruptive delay of permanent teeth in individuals with Ds and gubernacular canal obstruction through cone-beam computed tomography.

## 2. Materials and Methods

### 2.1. Study Population and Ethics Aspects

This observational, cross-sectional population-based study was conducted between January and July 2022 in Belém, the capital city of Pará State, in northern Brazil, and it included *n* = 15 individuals previously diagnosed with Ds who were spontaneously undergoing routine Ds treatment and management at the Integrated Service for Oral Diagnosis and Dental Care for Special Patients (SIDOPE).

The study complied with resolution 196/96 of the National Health Council and the Ethics and Research Committee of the Health of Science Institute from the Federal University of Pará (UFPA) and was approved under protocol number 3.405.106. Written informed consent was obtained from the parents or legal guardians of all participants prior to study enrollment.

### 2.2. Study Design

The sample consisted of underaged patients registered and treated at SIDOPE, 40 individuals and their parents or legal guardians were informed about the purpose of the study and invited to participate, although only 31 agreed to sign a written consent form before data collection and evaluation or met inclusion criteria. The study eligibility criteria were (i) individuals with a confirmed diagnosis of Ds; (ii) attended a monthly follow-up at SIDOPE in Pará State; (iii) at least 10 teeth in the mouth and without periodontal disease or cavities for about 1 year; (iv) aged between 10 to 15 years old; (v) resident in Pará State; (vi) both genders; and (vii) teeth completely encased within the alveolar bone.

The exclusion criteria were (i) individuals who were moved to other states; (ii) individuals who started follow-up at SIDOPE but no longer returned to monthly follow-up; (iii) medical records not filled out correctly; (iv) low-quality CBCT images; and (v) CBCT images with motion blurring or metal artifacts. Each participant was physically and orally evaluated in a private SIDOPE location. Clinical data were collected by a single researcher, a specialist in oral pathology, previously calibrated by Kappa test and with previous experience in clinical studies, and the intraoral clinical examination was performed in a dental office, dental chair, under indirect and artificial light, using a dental mirror, Williams periodontal probe (Hu-Friedy, Chicago, IL, USA) and clinical tweezers, all sterile.

After clinical exams, demographic and epidemiological data were obtained through a pretested standardized semi-structured questionnaire and medical records. The individuals were divided into two groups according to the presence or absence of Ds. The groups were formed and organized by nonsyndromic patients as the control group and Ds patients, with the nonsyndromic patients presenting with the GC open: G1 group: 16 patients (control group, GC open); G2 group: 15 (possible GC obstructed).

The study consisted of CBCT examinations for all 31 patients with impacted/unerupted permanent teeth in mixed dentition, for dental planning of orthodontic treatment, periodontal nonsurgical or surgical treatment, impacted teeth location, and identifying dental anomalies. All CBCT images were taken and collected from the Department of Oral and Maxillofacial Radiology of UFPA, the tomography machine was a Pax-i 3D smart (Green Vatech, Seoul, Korea), and acquisition parameters were a tube voltage of 95 kVp, a tube current of 7 mA, an exposure time of 5.9 s (Green Protocol) and voxel sizes and a field of view (FOV) of 0.15 mm (8 × 8 cm and 6 × 6 cm) and 0.30 mm (8 × 8 cm), respectively [19].

### 2.3. Imaging Procedures

After the CBCT images were acquired, they were converted to the digital imaging communication in medicine (DICOM) file format, uploaded to an independent storage software (Dropbox Inc., San Francisco, CA, USA, EUA), and then opened with open-source DICOM viewer software OsiriX 12.0 (Pixmeo, Geneva, Switzerland). All DICOM file evaluations were carried out by a single examiner, an oral radiologist with twenty years of experience, and a standardized CBCT view procedure (distance view, diagnostic screen, dimmed room, image magnification, contrast, brightness, and slice thickness) was established by the examiner [19].

All CBCT teeth images were visualized in axial, coronal, sagittal, cross-sectional, and panoramic views in the X, Y, and Z axes. The parameters analyzed included the presence or absence of GC, permanent eruption stages, the presence of physical obstructions in the eruption path or pathological conditions, root formation stage, angulation, and dilaceration [19]. Radiographically standard GC parameters were three-dimensional images with low-density, narrow diameter, corticated tract, and continuous with unerupted tooth dental follicle.

### 2.4. Statistical Analysis

The statistical analysis was performed using JAMOVI Statistics for Windows, Version 3 (Jamovi project, Sydney, Australia) statistical software. A descriptive analysis of the data was performed, with a distribution of absolute and relative frequencies, using the minimum and maximum value, mean and standard deviation as well as the *p*-value (*p* < 0.005) by G Test for the selected groups; additionally, the odds ratio (OR) and associated 95% confidence interval (CI) were used as measures of the strength of dependent association between GC absence (outcome) and Ds individuals.

## 3. Results

### 3.1. Sample Demographic and Epidemiological Characteristics

A total of 40 individuals were invited, although only 31 participated in the present study due to not satisfying the inclusion criteria or not agreeing to sign the consent term; therefore, the final analyzed sample consisted of 15 individuals diagnosed with Ds and 16 nonsyndromic individuals.

All clinical, demographic, and epidemiological characteristics are shown in Table 1. The average age was 12.51 (range: 10–15 years); regarding gender, males (*n* = 17—54.9%) were more prevalent than females (*n* = 14—45.1%); as self-declared ethnicity, white race was the most prevalent with 16 (51.6%) in both groups (G1: *n* = 6—37.5%)/G2: *n* = 10—66.6%); most of the individuals were from Belém city, capital of Pará State (*n* = 15—48.3%), in both groups (G1: *n* = 8—50%/ *n* = 7—46.6%); and regarding the type of education, most of the individuals attended private school (*n* = 21—67.7%) in both groups (G1: *n* = 10—62.5%/ *n* = 11—73.3%).

Regarding clinical disorders, the most common were blood disorders, hearing impairment, visual impairment, cardiovascular disorders, and endocrine disorders. Among blood disorders, the most prevalent was anemia (*n* = 8—33.3%), especially among Ds individuals (*n* = 7—46.6%), otitis media was the most prevalent hearing impairment (*n* = 12—50%), mainly in Ds patients (*n* = 10—66.6%), dacryocystitis and refractive errors were the most prevalent visual impairments (*n* = 18—75%), mostly in Ds patients (*n* = 12—80%), atrioventricular septal defect and systemic arterial hypertension were the most prevalent cardiovascular disorders (*n* = 20—83.3%), especially in Ds individuals (*n* = 14—93.3%), and prevalent endocrine disorders were the least common systemic conditions, although, among our sample, the most prevalent endocrine disorders were hypothyroidism (*n* = 7—29.2%) and mostly in Ds patients (*n* = 4—26.6%).

### 3.2. Clinical Oral Manifestations and Parafunctional Oral Habits

Regarding parafunctional oral habits (POH) among the individuals in this study, the most recurrent were digit sucking, tongue thrusting, bruxism, mouth breathing, nail biting, and obstructive sleep apnea. Bruxism was the most prevalent POH (*n* = 13/31—42%) in both groups (G1: *n* = 6/13—46.2%/ G2: *n* = 7/13—53.8%) followed by mouth breathing (*n* = 8/31–25.8%), digit sucking (*n* = 4/31—13%), tongue thrusting and obstructive sleep apnea (*n* = 3/31—9.6%), and during the analysis, no cases of nail biting were presented (Table 2).

Regarding oral manifestations (OM), hypodontia or missing teeth (not by dental caries) was the most prevalent OM for individuals (*n* = 17/70—24.2%) in both groups (G1: *n* = 6/17—35.5%/G2: *n* = 11/17—64.7%), followed by periodontal disease (*n* = 13/70—18.5%) in both groups (G1: *n* = 3/13—23%/G2: *n* = 10/13—77%), and maxillary atresia (*n* = 10/70—14.2%) in both groups (G1: *n* = 2/10—20%/G2: *n* = 8/10—80%), which are, according to the literature, common OMs for individuals with Ds.

### 3.3. Dental Prevalence and Teeth Eruption Disturbances

Through CBCT analysis, Table 3 shows the presence of permanent teeth and eruption state, which might be normal (*n* = 280—45.3%), impacted (*n* = 203—32.8%), or delayed/unerupted (*n* = 135—21.9%). Among normal, impacted, or delayed/unerupted teeth eruption states, there were a total of 618 teeth in 31 individuals, G1 had 351 (56.8%) teeth and G2 had 267 (43.2%). For dental prevalence, for both nonsyndromic and Ds patients, the most prevalent permanent teeth were maxillary central incisors (*n* = 85—13.7%), mandibular central incisors (*n* = 70—11.3%), maxillary lateral incisors (*n* = 63—10.2%), maxillary canines (*n* = 55—8.9%), and maxillary 1st molars (*n* = 54—10.8%). Regarding teeth eruption state, teeth with normal eruption were 280/618 (45.3%), G1 had 181/280 (64.6%) and G2 had 99/280 (35.4%), impacted teeth were 203/618 (32.8%), G1 had 64/203 (31.5%) and G2 had 139/203 (68.5%), and delayed/unerupted teeth were 135/618 (21.9%), G1 had 65/135 (48%) and G2 had 70/135 (52%).

Among teeth eruption state, the most common type of impacted permanent teeth were mandibular central incisors (*n* = 34—48.5%), maxillary canines (*n* = 28—50.9%), maxillary lateral incisors (*n* = 26—41.2%), mandibular lateral incisors (*n* = 24—47%), and maxillary central incisors (*n* = 22—25.8%). Regarding delayed/unerupted teeth, the most prevalent types were maxillary lateral incisors (*n* = 18—28.7%), maxillary canines (*n* = 16—29.1%), mandibular 1st molars (*n* = 14—26%), mandibular canines (*n* = 12—24.6%), maxillary 2nd molars (*n* = 11—44%), and maxillary central incisors (*n* = 11—12.5%). Evaluating both groups, G2 had the most prevalence of impacted or delayed/unerupted teeth among all individuals, and Ds craniofacial characteristics such as brachycephaly and OM such as maxillary atresia might influence the dental eruption state. Finally, a significant association between teeth eruption state and Ds was established for maxillary central incisors, mandibular central incisors, mandibular canines, and mandibular 2nd premolars (*p* < 0.0001).

### 3.4. Gubernacular Canal CBCT Detection

Among 618 teeth, the GC prevalence was 76.8% (*n* = 475), as shown in Table 4, and the GC structure, length, and diameter vary through dental eruption state (normal, impacted, or delayed/unerupted). Among permanent dentition, G1 (*n* = 295—62.1%) prevalence of GC was higher than G2 (*n* = 180—37.9%), therefore, we could suggest that Ds or the craniofacial characteristics associated with Ds might be a risk factor for GC obstruction. The detection rates of the GC in G1 permanent dentition were maxillary central incisors (38 GC/40 teeth—95%), mandibular central incisors (33 GC/33 teeth—100%), maxillary lateral incisors (30 GC/31 teeth—96.7%), maxillary canines (28 GC/29 teeth—96.5%), and mandibular canines (26 GC/26 teeth—100%); maxillary 1st premolars, maxillary 2nd premolars, mandibular lateral incisors, mandibular 1st premolars, and mandibular 2nd premolars had 100% GC rates, although compared with the previous teeth, there were fewer teeth with which to diagnose GC.

In G2, permanent dentition GC rates were mandibular 1st molars (21 GC/25 teeth—84%), mandibular central incisors (29 GC/37 teeth—78.3%), maxillary 1st molars (25 GC/32 teeth—78.1%), maxillary 2nd molars (10 GC/13 teeth—76.9%), and mandibular lateral incisors (23 GC/31 teeth—74.2%). Therefore, in analyzing the GC rates between groups, it appears that G2 had a decreased prevalence over G1, showing that in our sample, Ds itself, or Ds craniofacial characteristics, or OM could influence GC formation because in G1, there were 56 nondetected GCs and G2 had 87 nondetected GCs. Regarding a possible correlation between Ds individuals and lower GC detection rates, a bivariate logistic analysis identified a significant association in mainly mandibular canines (GC absence 44 times more likely in G2), maxillary canines (GC absence 27 times more likely in G2), mandibular central incisors (GC absence 22 times more likely in G2), and maxillary 1st premolars (GC absence 21 times more likely in G2).

Table 5 shows the GC prevalence rates among the different teeth formation and angulation statuses, and according to each group’s characteristics, in G1 normal teeth, formation and angulation had the highest GC prevalence of 120/295 (40.6%), impacted had 108/205 (36.6%), and delayed/unerupted 67/295 (22.8%). Significant differences in GC detection between groups were found, regarding G1 teeth formation status, root formation (72/120—60%) had higher GC detected rates than crown formation (48/120—40%), although in G2 crown formation, (67/180—83.7%) had higher GC detected rates than root formation, and among impacted teeth, GC detected rates were both higher in crown formation. In teeth formation analysis, all groups that had closed root apex during root formation had increased GC detected rates compared to opened root apex. Concerning teeth normal angulation, both groups had a higher prevalence of GC detection in normal, impacted, and delayed/unerupted, and no relevant association between dental formation status or teeth angulation and GC absence could be made.

### 3.5. Distribution Eruption Disturbances and Pathological Conditions

Regarding possible eruption disturbances (213/338—63%) or pathological conditions (125/338—37%) leading to GC detection, Table 6 shows all conditions. Among impacted and delayed/unerupted, there were 338/618 (54.7%) teeth, and the most prevalent eruption disturbances were maxillary or mandibular atresia (55/213—25.8%), followed by teeth ankyloses (42/213—19.7%) and teeth migration (39/213—18.3%). Additionally, eruption disturbances were more prevalent in G2 (119/213—57%) than in G1 (94/213—72.8%), especially maxillary or mandibular atresia (39/119—32.7%), which is one of the main Ds OM. Regarding the presence of pathological conditions, odontoma compound or complex (72/125—57.6%), supernumerary teeth (40/125—32%), and cystic lesions (11/125—8.8%), again being more prevalent in G2 (odontoma compound or complex: 52/90—55.7% and supernumerary teeth: 30/90—35.5%) than G1 (odontoma compound or complex: 10/35—28.5% and supernumerary teeth: 20/35—57.1%).

Concerning a viable correlation between eruption disturbances or pathological conditions related to Ds and lower GC detection rates, a bivariate logistic analysis identified a significant association between eruption disturbances and pathological conditions. According to our analysis, G2 was 0.5 times more likely to have eruption disturbance than G1, and G2 was twice more likely to have pathological conditions than G1. In pathological conditions, no significant association among each pathology was found, although, in eruption disturbances, there was a significant association between maxillary or mandibular atresia (2.4 times more likely to show in G2), teeth ectopic position (2.2 times more likely to show in G2), and follicle space widening (0.03 times more likely to show in G2). These results could demonstrate that Ds OM or craniofacial characteristics might delay or impact dental eruption state and also prejudice GC formation.

## 4. Discussion

To the best of the authors’ knowledge, the present study is one of the few studies that have focused on the use of CBCT imaging to study GC characteristics and its role in teeth eruption associated with pathological conditions among Ds individuals and is the first study evaluating this population in northern Brazil. According to Kaplan et al. [20], the first citation of gubernacular cord and GC was made in 1778 by John Hunter and was supported in 1887 by Charles Malassez in a microscopic evaluation. The overall concept of GC details that it is a radiolucent/hypodense corticated structure, directly connected to the dental follicle space, which is an eruption pathway for permanent teeth.

Through the years, studies demonstrated that GC structures can be visualized on panoramic and dental radiographs, although some diagnoses propose that the two-dimensional characteristics of dental radiographs would make it difficult to visualize or measure some GC aspects such as length, diameter, pathway, deformation, and/or obliteration. Therefore, recent studies introduced CBCT as a justified imaging examination to evaluate GC presence and its possible disturbances such as impacted or delayed/unerupted teeth that will influence clinical pediatric dentistry, orthodontics planning, and oral and maxillofacial surgical treatment due to its specificity in different images [21,22,23].

In our study, there were 143 nondetected GC among 618 teeth, and of these, there were 338 impacted or delayed/unerupted, mainly in G2 (*n* = 209/338—61.9%), which had, as major pathological conditions (*n* = 125/338—37%), odontoma in 72 (57.6%) cases, and supernumerary teeth in 40 (32%) cases. Additionally, we had a Ds individuals group in our study for which we evaluated common eruption disturbances (*n* = 213/338—63%) associated with this syndrome, which had a major influence on our 338 impacted or delayed/unerupted teeth [5,7,19,20]. According to the literature, GC absence or anatomical deformation could be possibly associated with various obstructive odontogenic tumors, such as compound or complex odontomas, mesiodens, or adenomatoid odontogenic tumors, which is demonstrated in our results [21,22,23,24,25,26,27,28,29,30,31].

In the general literature, some studies such as Koc et al. [19] and Nishida et al. [24] correlated GC detection and its prevalence with chronological age, and in confirming their results, a correlation between these parameters could not be found. Nishida et al. [24] found a higher prevalence of GC, and GC was clearly visualized on CBCT in patients even with unerupted permanent teeth during a normal eruption, except when maxillary central supernumerary teeth were present causing an eruption disturbance, which will obstruct GC; Koc et al. [19] had similarly low GC detection rate results to Nishida et al. [24] when eruption disturbances, as supernumerary teeth, were present. This study made a different correlation with sociodemographic data, to understand if other variables other than age could influence GC formation and prevalence, although no association was found.

Further, among studies that correlated GC detection and its prevalence with age, Oda et al. [22] identified that while patients were aging, the GC shape was modifying with chronological age, and according to Oda et al. [22], the progression forms of GC defined by authors were sprouting form groove form, imperfect-tubular form, tubular form, and hole form. Unfortunately, in our study, the first limitation we had was that we selected only CBCT images that did not contain abnormal GC shape findings; therefore, no correlation to GC shape could be made [21].

Of all the few studies regarding GC detection rates, our study was the first to evaluate a Ds population. The main reason why this sample was selected was that individuals with this genetic disorder may present various oral cavity alterations and specific oral/craniofacial characteristics such as class III malocclusions, maxillary hypoplasia, periodontal disease, dental caries, missing teeth, and some dental abnormalities such as microdontia, taurodontism, impacted teeth or hypodontia, and tongue disorders such as macroglossia that will affect their oral and respiratory functions, decreasing their oral quality of life [15,16,17,18]. Furthermore, due to these oral cavity alterations and Ds craniofacial characteristics, some parafunctional habits can be seen in these individuals such as tongue thrusting, bruxism, mouth breathing, and obstructive sleep apnea [25,26,27,28].

In our study, OM and parafunctional habits in both groups were evaluated and as seen in our results, a total of 70 OM were found, G2 (*n* = 49/70—70%) had a higher prevalence of OM than G1 (*n* = 21/70—30%), and the main OM were hypodontia or missing teeth, periodontal disease and maxillary atresia, all common OM in Ds. Another interesting fact in our results that corroborated the literature [27] is that there was a lower prevalence of dental caries (*n* = 5/70—7.1%) in G2 (*n* = 1/5—20%) than in G1 (*n* = 4/5—80%), whereas periodontal disease (13/70—18.5%) had a higher prevalence in G2 (*n* = 10/13—77%) than in G1 (*n* = 3/13—23%). Some parafunctional habits including bruxism, mouth breathing, digit sucking, tongue thrusting, and obstructive sleep apnea were observed; in total, there were 31 abnormal habits among both groups which are more prevalent in G2 (*n* = 18/31—58%) than in G1 (*n* = 13/31—42%).

Regarding the dental eruption state, there were 280 (45.3%) normal positioned teeth, 203 (32.8%) impacted teeth and 135 (21.9%) delayed/unerupted in a total of 618 teeth. In our results, G2 (*n* = 139/203—52%) had a higher prevalence of impacted teeth than G1 (64/203—18.2%), and the most common impacted permanent teeth were mandibular central incisors, maxillary canines, and maxillary lateral incisors. Oddly, delayed/unerupted teeth were most prevalent in G1 (*n* = 106/135—30%) compared to G2 (*n* = 29/135—11%), and the most common delayed/unerupted permanent teeth were maxillary lateral incisors, maxillary canines, and mandibular 1st molars. Furthermore, the only tooth which was statistically significant was the mandibular 2nd premolar [29,30,31]. This study had a few limitations that should be considered, including the small sample numbers from a cross-sectional analysis from a single institution, the fact that GC shape disturbances during teeth eruption or impaction could not be evaluated, and lastly, only individuals living in Pará State, northern Brazil, were included.

## 5. Conclusions

Based on the findings of our study, we concluded that GC was more common among nonsyndromic individuals than Ds individuals, the prevalence rates between the two groups presented a possible influence of Ds and its oral manifestations or craniofacial characteristics, which might increase the delay or impaction of teeth or dental eruption state, which will also prejudice GC formation, as demonstrated in our results. Therefore, the higher GC absence among Ds individuals may be a new disturbance eruption or impaction for these patients, so Ds might be a prevalence parameter for GC.

## Figures and Tables

**Table 1 jcm-12-03420-t001:** Individuals’ sociodemographic and epidemiological data.

Parameters	Total *n* = 31 (100%)	G1: Nonsyndromic *n* = 16 (51.6%)	G2: Down’s Syndrome *n* = 15 (48.4%)	*p*-Value *
Gender				
Female	14 (45.1%)	9 (56.2%)	5 (33.4%)	0.3561
Male	17 (54.9%)	7 (43.8%)	10 (66.6%)	
Ethnicity ^†^				
White	16 (51.6%)	6 (37.5%)	10 (66.6%)	0.2724
Black	5 (16.1%)	3 (18.7%)	2 (13.3%)	
Mixed	8 (25.8%)	5 (31.2%)	3 (20.1%)	
Indigenous	2 (6.4%)	2 (12.6%)	-	
Age (years)				
10–12	17 (54.8%)	12 (75%)	5 (33.4%)	0.0209
13–15	14 (45.2%)	4 (25%)	10 (66.6%)	
Source				
Belém	15 (48.3%)	8 (50%)	7 (46.6%)	0.1634
Ananindeua	8 (25.8%)	2 (12.5%)	6 (40%)	
Marituba	4 (13.1%)	2 (12.5%)	2 (13.4%)	
Santa Bárbara	2 (6.4%)	2 (12.5%)	-	
Benevides	2 (6.4%)	2 (12.5%)	-	
Type of education				
Private school	21 (67.7%)	10 (62.5%)	11 (73.3%)	0.7944
Public school	10 (32.3%)	6 (37.5%)	4 (26.7%)	
Blood disorders				
Yes	13 (42%)	6 (37.5%)	7 (46.6%)	0.8786
No	18 (58%)	10 (62.5%)	8 (53.4%)	
Hearing impairment				
Yes	14 (45.1%)	4 (25%)	10 (66.6%)	0.0209
No	17 (54.9%)	12 (75%)	5 (33.4%)	
Visual impairment				
Yes	23 (75%)	11 (68.7%)	12 (80%)	0.7603
No	8 (25%)	5 (31.3%)	3 (20%)	
Cardiovascular disorders				
Yes	25 (83.3%)	11 (68.7%)	14 (93.3%)	0.1926
No	6 (16.7%)	5 (31.3%)	1 (6.7%)	
Endocrine disorders				
Yes	8 (29.2%)	3 (25%)	4 (26.6%)	0.677
No	23 (70.8%)	12 (75%)	11 (73.4%)	

* G test; ^†^ Self-declared.

**Table 2 jcm-12-03420-t002:** Oral manifestations and parafunctional oral habits.

Parameters	Total *n* = 31 (100%)	G1: Nonsyndromic *n* = 16 (51.6%)	G2: Down’s Syndrome *n*= 15 (48.4%)	*p*-Value **
Parafunctional habits				0.2971
Digit sucking	4 (13%)	3 (75%)	1 (25%)	
Tongue thrusting	3 (9.6%)	-	3 (100%)	
Bruxism	13 (42%)	6 (46.2%)	7 (53.8%)	
Mouth breathing	8 (25.8%)	3 (37.5%)	5 (62.5%)	
Nail biting	0 (0%)	-	-	
Obstructive sleep apnea	3 (9.6%)	1 (33.4%)	2 (66.6%)	
Oral manifestations ^†^				0.1581
Dental caries	5 (7.1%)	4 (80%)	1 (20%)	
Periodontal disease	13 (18.5%)	3 (23%)	10 (77%)	
Malocclusion	9 (12.8%)	3 (33.4%)	6 (66.6%)	
Maxillary atresia	10 (14.2%)	2 (20%)	8 (80%)	
Drooling	7 (10%)	1 (14.3%)	6 (85.7%)	
Fissured tongue	4 (6.1%)	2 (50%)	2 (50%)	
Macroglossia	5 (7.1%)	-	5 (100%)	
Hypodontia or missing teeth *	17 (24.2%)	6 (35.3%)	11 (64.7%)	

* not by dental caries, ** G test, ^†^ multiple choice.

**Table 3 jcm-12-03420-t003:** Permanent teeth dental eruption state.

Parameters	Total Permanent Teeth *n* = 618 (100%)	G1: Nonsyndromic *n* = 351 (56.8%)	G2: Down’s Syndrome *n* = 267 (43.2%)	*p*-Value *
Maxillary central incisor	85 (13.7%)	40 (47%)	45 (53%)	<0.0001
Normal	52 (61.7%)	32 (80%)	20 (44.4%)	
Impacted	22 (25.8%)	2 (5%)	20 (44.4%)	
Delayed/unerupted	11 (12.5%)	6 (15%)	5 (11.2%)	
Maxillary lateral incisor	63 (10.2%)	31 (49.2%)	32 (50.8%)	0.2212
Normal	19 (30.1%)	8 (25.9%)	11 (34.3%)	
Impacted	26 (41.2%)	11 (35.4%)	15 (46.8%)	
Delayed/unerupted	18 (28.7%)	12 (38.7%)	6 (18.9%)	
Maxillary canine	55 (8.9%)	29 (52.7%)	26 (47.3%)	0.6975
Normal	11 (20%)	6 (20.6%)	5 (19.2%)	
Impacted	28 (50.9%)	16 (55.1%)	12 (46.1%)	
Delayed/unerupted	16 (29.1%)	7 (24.3%)	9 (34.7%)	
Maxillary 1st premolar	21 (3.4%)	11 (52.4%)	10 (47.6%)	0.0002
Normal	9 (42.8%)	9 (81.8%)	-	
Impacted	8 (38.1%)	1 (9.1%)	7 (70%)	
Delayed/unerupted	4 (19.1%)	1 (9.1%)	3 (30%)	
Maxillary 2nd premolar	17 (2.7%)	11 (64.7%)	6 (35.3%)	0.0006
Normal	10 (58.8%)	10 (90%)	-	
Impacted	4 (23.5%)	1 (10%)	3 (50%)	
Delayed/unerupted	3 (17.7%)	-	3 (50%)	
Maxillary 1st molar	54 (10.8%)	22 (40.8%)	32 (59.2%)	0.0443
Normal	29 (53.7%)	10 (45.4%)	19 (59.3%)	
Impacted	15 (27.7%)	10 (45.4%)	5 (15.7%)	
Delayed/unerupted	10 (18.6%)	2 (9.2%)	8 (25%)	
Maxillary 2nd molar	25 (4%)	12 (48%)	13 (52%)	0.9748
Normal	14 (56%)	7 (58.3%)	7 (53.8%)	
Impacted	-	-	-	
Delayed/unerupted	11 (44%)	5 (41.7%)	6 (46.2%)	
Mandibular central incisor	70 (11.3%)	33 (47.1%)	37 (52.9%)	<0.0001
Normal	31 (44.2%)	25 (75.7%)	6 (16.2%)	
Impacted	34 (48.5%)	5 (15.1%)	29 (78.3%)	
Delayed/unerupted	5 (7.3%)	3 (9.2%)	2 (5.5%)	
Mandibular lateral incisor	51 (8.2%)	20 (39.3%)	31 (60.7%)	0.0826
Normal	18 (35.3%)	13 (65%)	5 (16.1%)	
Impacted	24 (47%)	6 (30%)	18 (58%)	
Delayed/unerupted	9 (17.7%)	1 (5%)	8 (25.9%)	
Mandibular canine	49 (7.9%)	26 (53%)	23 (47%)	<0.0001
Normal	21 (42.8%)	16 (61.5%)	5 (21.7%)	
Impacted	16 (32.6%)	1 (3.9%)	15 (65.2%)	
Delayed/unerupted	12 (24.6%)	9 (34.6%)	3 (13.1%)	
Mandibular 1st premolar	18 (2.9%)	13 (72.2%)	5 (27.8%)	0.0053
Normal	6 (33.3%)	3 (23%)	3 (60%)	
Impacted	2 (11.1%)	-	2 (40%)	
Delayed/unerupted	10 (55.6%)	10 (77%)	-	
Mandibular 2nd premolar	23 (3.7%)	12 (52.1%)	11 (47.9%)	<0.0001
Normal	13 (56.5%)	12 (100%)	1 (9.2%)	
Impacted	4 (17.4%)	-	4 (36.3%)	
Delayed/unerupted	6 (26.1%)	-	6 (54.5%)	
Mandibular 1st molar	54 (8.7%)	29 (53.7%)	25 (46.3%)	0.1615
Normal	27 (50%)	17 (58.6%)	10 (40%)	
Impacted	13 (24%)	4 (13.9%)	9 (36%)	
Delayed/unerupted	14 (26%)	8 (27.5%)	6 (24%)	
Mandibular 2nd molar	33 (3.6%)	21 (63.6%)	12 (36.4%)	0.0042
Normal	20 (60.6%)	13 (61.9%)	7 (58.3%)	
Impacted	7 (21.2%)	7 (33.3%)	-	
Delayed/unerupted	6 (18.2%)	1 (4.8%)	5 (41.7%)	

* G test.

**Table 4 jcm-12-03420-t004:** Gubernacular canal prevalence detection.

Parameters	Total *n* = 475 (100%)	G1: Nonsyndromic *n* = 295 (62.1%)	G2: Down’s Syndrome *n* = 180 (37.9%)	*p*-Value **	Odds Ratio (95% CI)
		Detection Rates			
Maxillary central incisor	70 ^†^/85 * (82.3%)	38 ^†^/40 ^‡^ (95%)	32 ^†^/45 ^‡^ (71.1%)	0.0264	5.7778
Maxillary lateral incisor	51 ^†^/63 * (81%)	30 ^†^/31 ^‡^ (96.7%)	21 ^†^/32 ^‡^ (65.6%)	0.0276	10.6563
Maxillary canine	33 ^†^/55 * (60%)	28 ^†^/29 ^‡^ (96.5%)	5 ^†^/26 ^‡^ (19.2%)	0.0018	26.7692
Maxillary 1st premolar	12 ^†^/21 * (57.1%)	11 ^†^/11 ^‡^ (100%)	1 ^†^/10 ^‡^ (10%)	0.0448	20.8095
Maxillary 2nd premolar	13 ^†^/17 * (76.4%)	11 ^†^/11 ^‡^ (100%)	2 ^†^/6 ^‡^ (33.3%)	-	-
Maxillary 1st molar	43 ^†^/54 * (79.6%)	18 ^†^/22 ^‡^ (81.8%)	25/32 ^‡^ (78.1%)	-	-
Maxillary 2nd molar	20 ^†^/25 * (80%)	10 ^†^/12 ^‡^ (83.3%)	10/13 ^‡^ (76.9%)	0.7441	1.3846
Mandibular central incisor	62 ^†^/70 * (88.5%)	33 ^†^/33 ^‡^ (100%)	29/37 ^‡^ (78.3%)	0.0336	22.3333
Mandibular lateral incisor	43 ^†^/51 * (84.3%)	20 ^†^/20 ^‡^ (100%)	23/31 ^‡^ (74.2%)	0.1050	11.0635
Mandibular canine	30 ^†^/49 * (61.2%)	26 ^†^/26 ^‡^ (100%)	4/23 ^‡^ (13.4%)	0.0096	43.9787
Mandibular 1st premolar	14 ^†^/18 * (77.7%)	13 ^†^/13 ^‡^ (100%)	1/5 ^‡^ (20%)	0.0493	22.0909
Mandibular 2nd premolar	12 ^†^/23 * (52.1%)	12 ^†^/12 ^‡^ (100%)	-	-	-
Mandibular 1st molar	47 ^†^/54 * (87%)	26 ^†^/29 ^‡^ (89.6%)	21/25 ^‡^ (84%)	0.5908	1.5467
Mandibular 2nd molar	25 ^†^/33 * (75.7%)	19 ^†^/21 ^‡^ (90.4%)	6/12 ^‡^ (50%)	0.0634	5.2500

* Total teeth prevalence, ^‡^ total teeth prevalence per group, ^†^ total GC prevalence per teeth, ** G test.

**Table 5 jcm-12-03420-t005:** Gubernacular canal prevalence detection.

Parameters	G1: Nonsyndromic *n* = 295 (62.1%)		G2: Down’s Syndrome *n* = 180 (37.9%)		*p*-Value *
	GC Detected	GC Not Detected	GC Detected	GC Not Detected	
Teeth formation status					
Normal	120 (40.6%)	30 (53.5%)	80 (44.4%)	50 (57.4%)	0.0010
Crown formation	48 (40%)	20 (66.6%)	67 (83.7%)	40 (80%)	0.3558
Root formation	72 (60%)	10 (33.4%)	13 (16.3%)	10 (20%)	0.0039
Opened root apex	25 (34.8%)	2 (20%)	5 (38.5%)	5 (50%)	0.0193
Closed root apex	47 (65.2%)	8 (80%)	8 (61.5%)	5 (50%)	0.1353
Impacted	108 (36.6%)	20 (35.7%)	59 (32.7%)	16 (18.4%)	0.4060
Crown formation	68 (62.9%)	12 (60%)	36 (61%)	10 (62.5%)	0.9838
Root formation	40 (37.1%)	8 (40%)	23 (39%)	6 (37.5%)	0.8910
Opened root apex	15 (37.5%)	4 (50%)	9 (39.2%)	2 (33.4%)	0.7777
Closed root apex	25 (62.5%)	4 (50%)	14 (60.8%)	4 (66.6%)	0.7293
Delayed/unerupted	67 (22.8%)	6 (10.8%)	41 (22.9%)	21 (24.2%)	0.0004
Crown formation	-	-	-	-	-
Root formation	67 (100%)	6 (100%)	41 (100%)	21 (100%)	0.0004
Opened root apex	23 (34.4%)	1 (16.7%)	16 (39%)	7 (33.4%)	0.0387
Closed root apex	44 (65.6%)	5 (83.3%)	25 (61%)	14 (66.6%)	0.0078
Teeth angulation status					
Normal	120 (40.6%)	30 (53.5%)	80 (44.4%)	50 (57.4%)	0.0010
Normal	78 (65%)	20 (66.6%)	57 (71.2%)	25 (50%)	0.1674
Angulated	30 (25%)	8 (26.6%)	18 (22.5%)	19 (38%)	0.1674
Horizontal	12 (10%)	2 (6.8%)	5 (6.3%)	6 (12%)	0.0856
Inverted	-	-	-	-	-
Impacted	108 (36.6%)	20 (35.7%)	59 (32.7%)	16 (18.4%)	0.4060
Normal	65 (60.2%)	10 (50%)	28 (47.4%)	9 (56.2%)	0.2427
Angulated	30 (27.7%)	5 (25%)	17 (28.8%)	6 (37.5%)	0.4396
Horizontal	10 (9.2%)	5 (25%)	13 (22%)	1 (6.3%)	0.1911
Inverted	3 (2.9%)	-	1 (1.8%)	-	-
Delayed/unerupted	67 (22.8%)	6 (10.8%)	41 (22.9%)	21 (24.2%)	0.0004
Normal	39 (58.2%)	4 (66.6%)	25 (61%)	15 (71.4%)	0.0045
Angulated	22 (32.8%)	1 (16.7%)	13 (31.7%)	5 (23.8%)	0.0941
Horizontal	6 (9%)	1 (16.7%)	3 (7.3%)	1 (4.8%)	0.7055
Inverted	-	-	-	-	-

* G test.

**Table 6 jcm-12-03420-t006:** Prevalence of eruption disturbances and pathological condition among impacted and delayed/unerupted.

Parameters	Total *n* = 338 (100%)	G1: Nonsyndromic *n* = 129 (38.1%)	G2: Down’s Syndrome *n* = 209 (61.9%)	*p*-Value *	Odds Ratio (95% CI)
Eruption disturbances	213 (63%)	94 (72.8%)	119 (57%)	0.0035	0.4923
Maxillary or mandibular atresia	55 (25.8%)	16 (17%)	39 (32.7%)	0.0102	2.3766
Teeth ankyloses	42 (19.7%)	22 (23.4%)	20 (16.8%)	0.4100	1.2893
Teeth ectopic position	28 (13.1%)	13 (13.8%)	15 (12.6%)	0.0210	2.1750
Teeth migration	39 (18.3%)	17 (18%)	22 (18.4%)	0.9399	1.0273
Root dilaceration	21 (9.8%)	6 (6.4%)	15 (12.6%)	0.1374	2.1154
Third molar angulation	1 (0.7%)	1 (1.3%)	-	0.4122	0.2608
Adjacent root resorption	15 (7%)	7 (7.4%)	8 (6.9%)	0.8376	0.8958
Follicle space widening	12 (5.6%)	12 (12.7%)	-	0.0133	0.0276
Pathological conditions	125 (37%)	35 (27.2%)	90 (43%)	0.0035	2.0312
Cystic lesions	11 (8.8%)	3 (8.5%)	8 (8.8%)	0.9551	1.0407
Odontogenic fibroma	2 (1.6%)	2 (5.9%)	-	0.0957	0.0740
Supernumerary teeth	40 (32%)	10 (28.5%)	30 (35.5%)	0.6087	1.2500
Odontoma ^†^	72 (57.6%)	20 (57.1%)	52 (55.7%)	0.9486	1.0263

* G test, ^†^ compound or complex.

## Data Availability

All data referred to in this study are available in the manuscript.

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
