# Peer review of "CBCT Assessment of Gubernacular Canals on Permanent Tooth Eruption in Down’s Syndrome"

_jcm, 2023, doi:10.3390/jcm12103420_

Round 1
Reviewer 1 Report
The manuscript proposes to investigate the correlation of the delayed eruption of permanent teeth in individuals with Down’s syndrome and gubernacular canal using cone-beam computed tomography imaging.
The authors nicely outlined the manuscript , however major revision of language and writing styles is recommended due to substantial errors in translation from Portuguese to English that weakens the manuscript.
Another weakness is that in the results section the authors could better explain the number they are providing in the tables . eg: total number of teeth present vs total number of unerupted and impacted teeth because it was very confusing to understand the prevalence calculation.
Comments on the manuscript are outlined below:
1) Some examples of problems with the language aspects are noted in the paragraph - line 50-56 - reconsider re-writing this paragraph _ eg: dental lamina instead of teeth lamina; the word "substances" when referring to cells , growth factors, etc ; use of the word "prejudice" instead of delayed or affected
2) Table 3 ) consider adding on table3 the word permanent teeth over the total n=618
3) Is 618 teeth the total number of permanent teeth present or total number of permanent teeth that are still unerupted/ impacted ? please clarify in the manuscript.
3) There is a big difference when looking at the total number of permanent teeth comparing the control with the DS patients ( 351 vs 267 ) which suggests that Down syndrome patients had more missing teeth compared to the control. Would that difference interfere in the lower prevalence of GC?
4) It was not clear to me how prevalence was calculated because is my understanding that the 618 teeth analyzed there were 475 teeth that presented GC. please expand on what radiographic features would determine the presence of GC.
5) Line 218 - "Among permanent dentition G1 (n=295 - 62.1%) was more prevalent than G2 (n=180 - 37.9%)" Consider a changing "Among permanent dentition, G1 (n=295 - 62.1%) prevalence of GC was higher than G2 (n=180 - 37.9%)"
6) The numbers are very confusing because in table 5 : gubernacular canal prevalence detection the total numbers of G1 and G2 ( 295 vs 180) is then divided into GC detected or not detected , so I don't understand how GC prevalence was calculated.
Author Response
Reply to reviewer #1
1. Concern of the reviewer • Some examples of problems with the language aspects are noted in the paragraph - line 50-56 - reconsider re-writing this paragraph _ eg: dental lamina instead of teeth lamina; the word "substances" when referring to cells , growth factors, etc ; use of the word "prejudice" instead of delayed or affected.
Our response: Dear Reviewer #1, we appreciate your suggestion and text was carefully revised.
Revised text:Page 2, lines 50-56, “The gubernacular cord formation begins with the remnants cells of the dental lamina that are assembled as a fibrous cord through enamel organ’s reduced epithelium towards the oral mucosa [8-13]. In the gubernacular cord there are chemical inflammatory mediator, epithelial growth factor, osteoclasts, osteoblasts and during bone deremineralization process, a space is formed around gubernacular cord, which is denominated gubernacular canal (GC) [8-13]. However, in some cases physiological permanent teeth eruption process is delayed due to GC obstruction.”
2. Concern of the reviewer • Table 3 ) consider adding on table3 the word permanent teeth over the total n=618.
Our response: Dear Reviewer #1, we appreciate your concern and the table 3 text was carefully rewritten.
Revised text:Page 6, line 213, “Table 3. Permanent teeth dental eruption state.”
3. Concern of the reviewer • Is 618 teeth the total number of permanent teeth present or total number of permanent teeth that are still unerupted/ impacted ? please clarify in the manuscript.
Our response: Dear Reviewer #1, we appreciate your suggestion and the text was carefully explained.
Revised text:Page 6, lines 192-193, “Among Normal; Impacted or Delayed/Unerupted teeth eruption state there were 618 teeth throughout 31 individuals.”
4. Concern of the reviewer • There is a big difference when looking at the total number of permanent teeth comparing the control with the DS patients ( 351 vs 267 ) which suggests that Down syndrome patients had more missing teeth compared to the control. Would that difference interfere in the lower prevalence of GC?.
Our response: Dear Reviewer #1, we appreciate your concern. So due to all Down’s syndrome craniofacial characteristics to our group there was a reasonable possibility that Down’s syndrome might be a possible prevalence factor to gubernacular canal presence or absence, therefore that is why we chosen to study with Down’s syndrome patients versus normal patients and according to our results Down’s syndrome proved to be a possible prevalence parameters to gubernacular canal, as newly stated in our conclusion. I hope to help you to understand why we choose Down’s syndrome patient to evaluate gubernacular canal.
5. Concern of the reviewer • It was not clear to me how prevalence was calculated because is my understanding that the 618 teeth analyzed there were 475 teeth that presented GC. please expand on what radiographic features would determine the presence of GC. It was not clear to me how prevalence was calculated because is my understanding that the 618 teeth analyzed there were 475 teeth that presented GC. please expand on what radiographic features would determine the presence of GC.
Our response: Dear Reviewer #1, we appreciate your suggestion and the text was carefully added.
Revised text:Page 3, lines 140-142, “Radiographically standard GC parameters were three-dimensional images with low-density, narrow diameter, corticated tract and continuously with unerupted tooth dental follicle.”
6. Concern of the reviewer • Line 218 - "Among permanent dentition G1 (n=295 - 62.1%) was more prevalent than G2 (n=180 - 37.9%)" Consider a changing "Among permanent dentition, G1 (n=295 - 62.1%) prevalence of GC was higher than G2 (n=180 - 37.9%)"
Our response: Dear Reviewer #1, we appreciate your suggestion and text was carefully revised.
Revised text:Page 3, lines 107-110, “Among permanent dentition, G1 (n=295 - 62.1%) prevalence of GC was higher than G2 (n=180 - 37.9%).”
7. Concern of the reviewer • The numbers are very confusing because in table 5 : gubernacular canal prevalence detection the total numbers of G1 and G2 ( 295 vs 180) is then divided into GC detected or not detected , so I don't understand how GC prevalence was calculated.
Our response: Dear Reviewer #1, we appreciate your concern, table 5 is an idea from our study based on other in literature to evaluate gubernacular canal prevalence detection among teeth during all different stages of dental formation (crown or root formation) or its angulation and position to see in which stage among groups the prevalence is lower and the not detected part is to see in which dental formation (crown or root formation) stage or its angulation and position the absence of gubernacular canal is higher and all data was calculated with G test and percentages. I hope to help you to understand this table and I would like to appreciate all your concerns, for sure it helped to improve the manuscript.

Reviewer 2 Report
This is a paper summarizing abnormal findings, including GC, in patients with Down syndrome. However, the data are inadequate to discuss the relationship between GC and other symptoms. It is true that there are no published data summarizing GC in Down syndrome. I believe that the paper, especially the discussion, should be revised as data summarizing abnormal findings in Down syndrome.
The phrases "Ds," "Down's Syndrome," and "Down's Syndrome" need to be unified. Also, is there any intention to use "G2" and "with Down's Syndrome" differently? They should be unified.
l.60 The original literature investigating the relationship between GC and dental disease is as follows
A spatial association between odontomas and the gubernaculum tracts. Oral Surg Oral Med Oral Pathol Oral Radiol 2016 Vol. 121 Issue 1 Page 91-5
Characteristics of the gubernaculum tracts in mesiodens and maxillary anterior teeth with delayed eruption on MDCT and CBCT. Oral Surg Oral Med Oral Pathol Oral Radiol 2016 Vol. 122 Issue 4 Pages 511-6
Significance and usefulness of imaging characteristics of gubernaculum tracts for the diagnosis of odontogenic tumors or cysts. PLoS One 2018 Vol. 13 Issue 7 Pages e0199285
These references need to be added to the bibliography.
Some sentences, in conclusion, should be written in the discussion (e.g., l.219, l.230).
There are some questions about the numbers in the data set.
l.152: l.152: Is not the number of individuals excluded 9, or is six correct?
Table 3: Shouldn't the totals for G1 and G2 be 310 and 308, respectively?
The table is confusing. The table headings in rows should not include percentages. Why not list the totals in the bottom row? Of the column headings, the dental type should be left-aligned and bold. I have no idea what the column headings are in Table 5.
What are the explanatory and objective variables in a bivariate logistic analysis? I have no idea what the odds ratio means. I guess you need to check with a statistician to ensure that the correct statistics are being done.
l.296: The studies on the Gubernacular canal (e.g., Refs. 21-23) have never recommended CBCT as a routine exam to evaluate the presence of GC. They performed those studies as retrospective studies. Those CT examinations were performed to diagnose disorders such as masses and embedded teeth.
l. 320-328: Ref. 22 is cited, but you should cite 21 should be cited in this paragraph.
l. 322: The article (Ref. 21) stated that the detection rate of canals continuing from the distal side of the mandibular third molar (Pseudo-GT as named by Oda et al.) is more common in 12-14 years of age. Pseudo-GT differs from GC in permanent teeth.
Author Response
Reply to reviewer #2
1. Concern of the reviewer • The phrases "Ds," "Down's Syndrome," and "Down's Syndrome" need to be unified. Also, is there any intention to use "G2" and "with Down's Syndrome" differently? They should be unified.
Our response: Dear Reviewer #2, we appreciate your concern. The text was carefully revised a manuscript long.
2. Concern of the reviewer• 60 The original literature investigating the relationship between GC and dental disease is as follows:A spatial association between odontomas and the gubernaculum tracts. Oral Surg Oral Med Oral Pathol Oral Radiol 2016 Vol. 121 Issue 1 Page 91-5;Characteristics of the gubernaculum tracts in mesiodens and maxillary anterior teeth with delayed eruption on MDCT and CBCT. Oral Surg Oral Med Oral Pathol Oral Radiol 2016 Vol. 122 Issue 4 Pages 511-6;Significance and usefulness of imaging characteristics of gubernaculum tracts for the diagnosis of odontogenic tumors or cysts. PLoS One 2018 Vol. 13 Issue 7 Pages e0199285.These references need to be added to the bibliography.
Our response: Dear Reviewer #2, we appreciate your concern. The references were carefully added.
Revised text:Page 14, lines 450-456, “Oda, M.; Miyamoto, I.; Nishida, I.; Tanaka, T.; Kito, S.; Seta, Y.; Yada, N.; et al. A spatial association between odontomas and the gubernaculum tracts. Oral Surg Oral Med Oral Pathol Oral Radiol 2016, 121(1):91-5.Oda, M.; Nishida, I.; Miyamoto, I.; Habu, M.; Yoshiga, D.; Kodama, M.; Osawa, K.; et al. Characteristics of the gubernaculum tracts in mesiodens and maxillary anterior teeth with delayed eruption on MDCT and CBCT. Oral Surg Oral Med Oral Pathol Oral Radiol 2016, 122(4):511-6.Oda, M.; Nishida, I.; Miyamoto, I.; Saeki, K.; Tanaka, T.; Kito, S.; Yamamoto, N.; et al. Significance and usefulness of imaging characteristics of gubernaculum tracts for the diagnosis of odontogenic tumors or cysts. PLoS One 2018, 13(7):e0199285. ”
3 Concern of the reviewer Some sentences, in conclusion, should be written in the discussion (e.g., l.219, l.230).
Our response: Dear Reviewer #2, we appreciate your suggestion, although we would like to maintain these sentences in conclusion to impact our findings to readers and highlight them.
4 Concern of the reviewer There are some questions about the numbers in the data set.
l.152: l.152: Is not the number of individuals excluded 9, or is six correct?
Table 3: Shouldn't the totals for G1 and G2 be 310 and 308, respectively?
Our response: Dear Reviewer #2, we appreciate your doubts. The text was carefully revised about question 1, to answer question 2 regarding table 3, no the correct totals are G1: n=351 and G2: n=267.
Revised text: Page 4, line 154, “nine.”
5 Concern of the reviewer The table is confusing. The table headings in rows should not include percentages. Why not list the totals in the bottom row? Of the column headings, the dental type should be left-aligned and bold. I have no idea what the column headings are in Table 5.
Our response: Dear Reviewer #2, we appreciate your kind doubt. Let me explain about table 5. Table 5 is an idea from our study based on other in literature to evaluate gubernacular canal prevalence detection among teeth during all different stages of dental formation (crown or root formation) or its angulation and position to see in which stage among groups the prevalence is lower and the not detected part is to see in which dental formation (crown or root formation) stage or its angulation and position the absence of gubernacular canal is higher. Regarding rows and percentages we appreciate so much your suggestion, although we all see total list in upper rows.
6 Concern of the reviewer What are the explanatory and objective variables in a bivariate logistic analysis? I have no idea what the odds ratio means. I guess you need to check with a statistician to ensure that the correct statistics are being done.
Our response: Dear Reviewer #2, we appreciate your concern. The explanatory and objective variables in a bivariate logistic analysis was suggested By Dr. Rogerio Valois whom is a Northern Brazilian renowned statistician, he explained that Binary Regression uses one or more predictor variables that may be continuous or categorical to predict if dependent variables will impact directly the main variable which in this case is gubernacular canal, then these relations will be influenced in either a negative or positive result.
7 Concern of the reviewer l.296: The studies on the Gubernacular canal (e.g., Refs. 21-23) have never recommended CBCT as a routine exam to evaluate the presence of GC. They performed those studies as retrospective studies. Those CT examinations were performed to diagnose disorders such as masses and embedded teeth.
Our response: Dear Reviewer #2, we appreciate your suggestion. The text was carefully revised.
Revised text:Page 11, lines 299-302, “Therefore, recently studies introduced CBCT as justified image exam to evaluate GC presence and its possible disturbances as impacted or delayed/unerupted teeth that will influence in clinically pediatric dentistry, orthodontics planning and oral and maxillofacial surgery treatment due to its specificity in different images [21-23].”
8 Concern of the reviewer l. 320-328: Ref. 22 is cited, but you should cite 21 should be cited in this paragraph.
Our response: Dear Reviewer #2, we appreciate your suggestion. The reference was carefully added.
Revised text:Page 12, line 331, “[21].”
9 Concern of the reviewer l. 322: The article (Ref. 21) stated that the detection rate of canals continuing from the distal side of the mandibular third molar (Pseudo-GT as named by Oda et al.) is more common in 12-14 years of age. Pseudo-GT differs from GC in permanent teeth.
Our response: Dear Reviewer #2, we appreciate your suggestion. The text was carefully revised.
Revised text:Page 12, lines 323-326, “Still among studies that correlated GC detection and its prevalence with age, Oda et al. [22] found a significant association between what authors called Pseudo- gubernacular canal shape alterations and chronological age, according to authors ages between 12 to 14 there was the highest prevalence of Pseudo- gubernacular canal.”

Round 2
Reviewer 2 Report
Thank you for your advice. With the table corrected, I now understand the value of this paper. However, there are still a few things that need to be corrected.
l. 26 “A total of 618 teeth among 31 individuals were analyzed, 475 (76.8%) GC were CBCT detected in 23/31 patients and of these 8 individuals of not detected GC, 6 belong to G2.”
It reads, "23 people had GCs found on all of their teeth and 8 people had no GCs found at all." Does it mean "23 had GCs found on all teeth and 8 had no GCs found on some teeth"?
l.67
The word "yet" appears twice. Please delete one of them.
Again, I still feel that the total in Table 3 is incorrect.
l.202 “Delayed/Unerupted teeth were 135 (21.9%), G1 had 106 (30%) and G2 had 29 (11%).” Is this correct? From Table 3, there were 135 Delayed/Unerupted teeth, but 65 for G1 and 70 for G2. The numbers in the text and Table 3 do not match.
If my interpretation is wrong, please explain. However, I think this is not clear to the reader as well as to me.
l.323-326
What I meant in my first review is that Pseudo-GT has nothing to do with this paper. I think this sentence should be deleted.
Ref. 29, 30, and 31 should be cited on line 312. (“According to literature, GC absence or anatomical deformation could be possibly associated to various obstructive odontogenic tumors as compound or complex odontomas, mesiodens or adenomatoid odontogenic tumor, which is demonstrated in our results”) Not on line 362.
Author Response
Reply to reviewer #2
1. Concern of the reviewer
• Thank you for your advice. With the table corrected, I now understand the value of this paper. However, there are still a few things that need to be corrected.
Our response: Dear Reviewer #2, we appreciate your kind correction in order to improve our manuscript, specially table 3 text was correct for better understanding, we hope now it is suitable for publication.
2. Concern of the reviewer
• l. 26 “A total of 618 teeth among 31 individuals were analyzed, 475 (76.8%) GC were CBCT detected in 23/31 patients and of these 8 individuals of not detected GC, 6 belong to G2.” It reads, "23 people had GCs found on all of their teeth and 8 people had no GCs found at all." Does it mean "23 had GCs found on all teeth and 8 had no GCs found on some teeth"?
Our response: Dear Reviewer #2, we appreciate your concern. The text was carefully revised.
Revised text:Page 1, lines 25-26, “A total of 618 teeth among 31 individuals were analyzed, 475 (76.8%) GC were detected by CBCT in 23/31 patients and of these 6 belong to G2.”
3. Concern of the reviewer
• The word "yet" appears twice. Please delete one of them.
Our response: Dear Reviewer #2, we appreciate your suggestion and the word “yet” was deleted.
Revised text:Page 2, lines 66-71, “GC obstruction among individuals with Ds has not yet been reported in the literature. A few studies demonstrated oral cavity abnormalities in individuals with Ds, except does not correlate with GC obstruction. de Moraes et al. [15] evaluated the incidence of dental anomalies in permanent dentition among individuals with Ds using panoramic x-rays and as results the authors observed dental anomalies in 50.47% of the cases and retained teeth or supernumerary teeth were between the most prevalent abnormalities.”
4. Concern of the reviewer
• Again, I still feel that the total in Table 3 is incorrect. l.202 “Delayed/Unerupted teeth were 135 (21.9%), G1 had 106 (30%) and G2 had 29 (11%).” Is this correct? From Table 3, there were 135 Delayed/Unerupted teeth, but 65 for G1 and 70 for G2. The numbers in the text and Table 3 do not match. If my interpretation is wrong, please explain. However, I think this is not clear to the reader as well as to me.
Our response: Dear Reviewer #2, we appreciate your doubts. The table 3 text was carefully revised upon your key observations, thanks for it and we hope now to improve your experience reading or paper.
Revised text: Page 6, lines 192-203, “Through CBCT analysis, table 3 show permanent teeth presence and eruption’s state, which might be Normal (n=280 – 45.3%), Impacted (n=203 – 32.8%) or Delayed/Unerupted (n=135 - 21.9%). Among Normal; Impacted or Delayed/Unerupted teeth eruption state there were 618 teeth throughout 31 individuals, G1 had 351 (56.8%) teeth and G2 had 267 (43.2%). Throughout non-syndromic and Ds patients dental prevalence, the most prevalent permanent teeth were maxillary central incisor (n=85 - 13.7%); mandibular central incisor (n=70 - 11.3%); maxillary lateral incisor (n=63 - 10.2%); maxillary canine (n=55 – 8.9%) and maxillary 1st molar (n=54 - 10.8%). In the matter of teeth eruption’s state, teeth with normal eruption were 280/618 (45.3%), G1 had 181/280 (64.6%) and G2 had 99/280 (35.4%). Impacted teeth were 203/618 (32.8%), G1 had 64/203 (31.5%) and G2 had 139/203 (68.5%). Delayed/Unerupted teeth were 135/618 (21.9%), G1 had 65/135 (48%) and G2 had 70/135 (52%).”
5. Concern of the reviewer
• What I meant in my first review is that Pseudo-GT has nothing to do with this paper. I think this sentence should be deleted.
Our response: Dear Reviewer #2, we appreciate your suggestion and the text was deleted.
Revised text: Page 12, lines 323-329, “Still among studies that correlated GC detection and its prevalence with age, Oda et al. [22] identified that while patients were aging the GC shape was modifying with chronological age and according to Oda et al. [22] the progression forms of GC defined by authors were sprouting form, groove form, imperfect‐tubular form, tubular form and hole form. Unfortunately, in our study the first limitations we had was we selected only CBCT images that contained no abnormal GC shape findings therefore no correlation to GC shape could be made [21].”
6. Concern of the reviewer
• Ref. 29, 30, and 31 should be cited on line 312. (“According to literature, GC absence or anatomical deformation could be possibly associated to various obstructive odontogenic tumors as compound or complex odontomas, mesiodens or adenomatoid odontogenic tumor, which is demonstrated in our results”) Not on line 362.
Our response: Dear Reviewer #2, we appreciate your concern. The text was carefully revised.
Revised text:Page 11, lines 309-312, “According to literature, GC absence or anatomical deformation could be possibly associated to various obstructive odontogenic tumors as compound or complex odontomas, mesiodens or adenomatoid odontogenic tumor, which is demonstrated in our results [21-31].”